# ZeroSCROLLS: A Zero-Shot Benchmark for Long Text Understanding

Uri Shaham[τ]    Maor Ivgi[τ]    Avia Efrat[τ]    Jonathan Berant[τ]    Omer Levy[τμ]

[τ] The Blavatnik School of Computer Science, Tel Aviv University
[μ] Meta AI

## Abstract

We introduce ZeroSCROLLS, a zero-shot benchmark for natural language understanding over long texts, which contains only test and small validation sets, without training data. We adapt six tasks from the SCROLLS benchmark, and add four new datasets, including two novel information fusing tasks, such as aggregating the percentage of positive reviews. Using ZeroSCROLLS, we conduct a comprehensive evaluation of both open-source and closed large language models, finding that Claude outperforms ChatGPT, and that GPT-4 achieves the highest average score. However, there is still room for improvement on multiple open challenges in ZeroSCROLLS, such as aggregation tasks, where models struggle to pass the naive baseline. As the state of the art is a moving target, we invite researchers to evaluate their ideas on the live ZeroSCROLLS leaderboard.[1]

## 1 Introduction

Large language models (LLMs) have been improving at an incredible pace, solving problems that seemed out of reach, without any task-specific training examples (Wei et al., 2022a; Ouyang et al., 2022; OpenAI, 2023). As commercial LLMs are adopted worldwide, it becomes clear that they must also operate successfully over long sequences, such as conversation histories or scientific documents. However, current LLM benchmarks that do evaluate models in a zero-shot setting, such as HELM (Liang et al., 2022) and BigBench (Srivastava et al., 2022), mostly focus on short sequences; BigBench, for example, has an average of 77 words per input. To fill this gap, we introduce ZeroSCROLLS: Zero-Shot CompaRison Over Long Language Sequences, a benchmark for zero-shot long text reasoning over natural language, and conduct a thorough investigation of state-of-the-art LLMs.

ZeroSCROLLS extends SCROLLS (Shaham et al., 2022), a long text understanding benchmark that

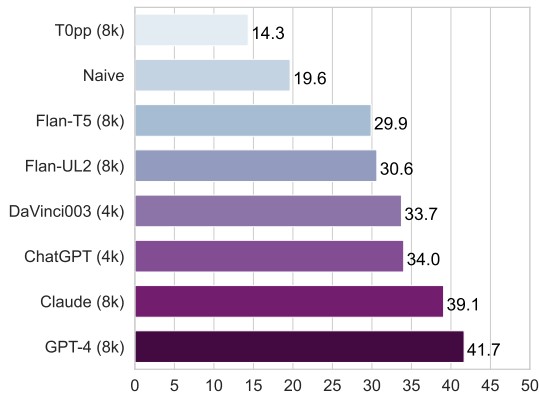

Figure 1: ZeroSCROLLS measures the average performance of state-of-the-art language models across 10 long text understanding tasks. The maximal amount of tokens each model can process is given in parentheses.

enables fine-tuning, adding four additional tasks: query-based summarization, multi-hop question answering, sentiment aggregation, and sorting book chapter summaries. We specifically design the latter two tasks to examine a model's ability to aggregate and compare information across long sequences, while keeping evaluation simple and accurate. ZeroSCROLLS is designed to test *zero-shot* capabilities, and contains test sets with simple natural prompts and private gold references, small validation sets, and no train data. It has a live leaderboard to enable transparent and dynamic progress. Figure 1 shows the state of the leaderboard based on our experiments, and Figure 2 shows a per-task breakdown of a selected subset of models.

We use this new testbed to perform extensive evaluation and analysis across state-of-the-art open and closed models. On question answering tasks, we find that zero-shot LLMs bridge the gap with task-specific fine-tuned models; GPT-4 sets a new state of the art on the challenging QuALITY task (Pang et al., 2022), almost reaching human performance. In contrast, LLMs generally struggle to

[1] https://www.zero.scrolls-benchmark.com/

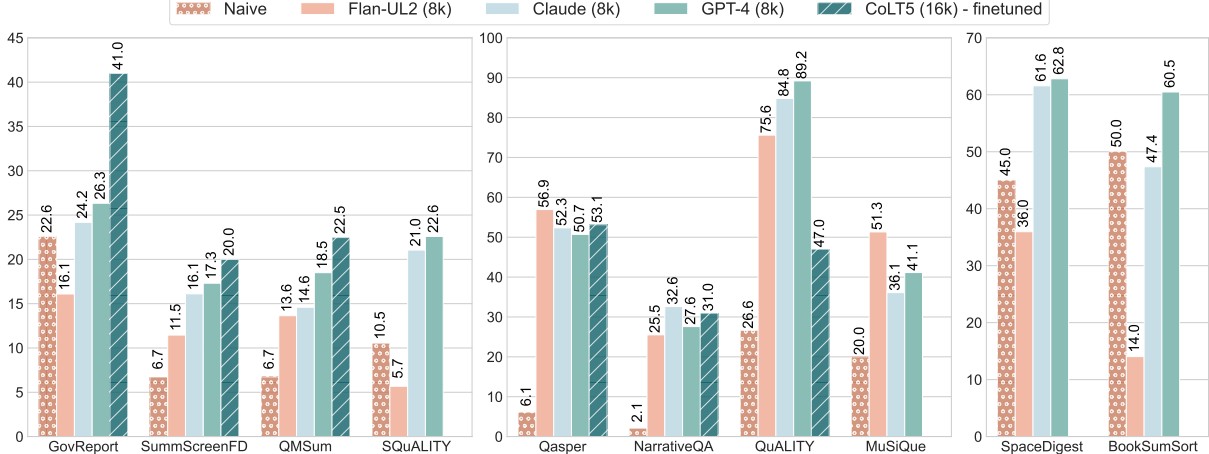

Figure 2: Per task scores of various LLMs and other baselines. In parentheses: the maximum number of tokens.

obtain such high scores for summarization tasks without a training set from which to learn the nuances and artifacts of each dataset, even though GPT-4 does approach the fine-tuned state of the art on two of three datasets. We also observe that two of our new tasks, sentiment aggregation and sorting book chapter summaries, prove exceptionally challenging for all LLMs, with only GPT-4 surpassing the naive baseline in each task. Our code is available online.[2]

When analyzing GPT-4 responses, we often find correct answers that do not match the requested format; e.g. producing a full sentence when asked to answer in a single phrase. This problem is not unique to GPT-4, as different models may deviate from the specified format in different tasks. While ZeroSCROLLS is primarily aimed at facilitating research in understanding long texts, we encourage the community to use this benchmark to advance research in instruction understanding, prompt engineering, and evaluation of generated texts as well.

## 2   Background: SCROLLS

SCROLLS (Shaham et al., 2022) was introduced as a long text understanding benchmark. Its datasets were curated, cleaned, and reformatted to a single input-output format allowing for easy and fast usage, with every example containing a single long document, such as a scientific paper or a book. Since its launch, SCROLLS has facilitated significant progress, including new pretraining objectives (Tay et al., 2023), adaptations of short-text models to long sequences (Phang et al., 2022; Xiong et al., 2022; Ivgi et al., 2023; Bertsch et al., 2023), and

dedicated long sequence models pretrained from scratch (Guo et al., 2022; Ainslie et al., 2023).

All the aforementioned methods eventually fine-tune a specialized model for every single task in SCROLLS, a setting that remains important for many applications. However, in the modern era of general purpose, zero-shot reasoning LLMs, a new evaluation setup is required, where this dependence on task-specific fine-tuning is alleviated.

## 3   The ZeroSCROLLS Benchmark

ZeroSCROLLS is a zero-shot benchmark containing test sets of ten natural language tasks, each one requiring reasoning over a different type of long text. To ensure affordability, we limit every task to a maximum of 500 examples.

### 3.1   Tasks

We describe the different ZeroSCROLLS datasets, six of which we adapt from Shaham et al. (2022), and four new tasks. Table 1 provides an overview.

#### 3.1.1   Summarization

We adopt the three summarization datasets from SCROLLS (GovReport, SummScreenFD, and QMSum), and add a fourth (SQuALITY). GovReport and SummScreenFD are full-document summarization tasks, while QMSum and SQuALITY are query-focused.

**GovReport**   (Huang et al., 2021) contains long reports by the Congressional Research Service and the U.S. Government Accountability Offices, with their expert written summaries.

---

[2]https://github.com/tau-nlp/zero_scrolls

| Dataset | Task | Domain | Metric | Avg #Words | #Examples |
|---|---|---|---|---|---|
| GovReport (Huang et al., 2021) | Summarization | Government | ROUGE | 7,273 | 500 |
| SummScreenFD (Chen et al., 2022) | Summarization | TV | ROUGE | 5,663 | 337 |
| QMSum (Zhong et al., 2021) | QB-Summ | Meetings | ROUGE | 10,839 | 281 |
| SQuALITY (Wang et al., 2022) | QB-Summ | Literature | ROUGE | 4,971 | 260 |
| Qasper (Dasigi et al., 2021) | QA | Science | F1 | 3,531 | 500 |
| NarrativeQA (Kočiský et al., 2018) | QA | Literature, Film | F1 | 49,384 | 500 |
| QuALITY (Pang et al., 2022) | MC-QA | Literature, Misc | Accuracy | 4,248 | 500 |
| MuSiQue (Trivedi et al., 2022) | QA | Wikipedia | F1 | 1,749 | 500 |
| SpaceDigest (New) | Aggregation | Reviews | ES | 5,481 | 500 |
| BookSumSort (New) | Aggregation | Literature | $C_{idx}$ | 6,840 | 500 |

Table 1: An overview of the data statistics in ZeroSCROLLS. *QB-Summ* means query-based summarization, *MC-QA* abbreviates multiple-choice question answering. *ES* refers to exponential similarity and $C_{idx}$ refers to concordance index. SpaceDigest data is from on the Space dataset (Angelidis et al., 2021) and BookSumSort data is from the BookSum dataset (Kryscinski et al., 2022).

**SummScreenFD** (Chen et al., 2022) contains episode scripts from TV shows with community contributed recaps that were collected from Wikipedia and TVMaze as their summaries.

**QMSum** (Zhong et al., 2021) is a query-based summarization dataset over meetings transcripts. It contains academic meetings, industrial product meetings, and Welsh and Canadian parliament transcripts. Alongside the meeting transcript, each instance contains a query, which aims to focus the summary on a particular topic.

**SQuALITY** (Wang et al., 2022) is a question-focused summarization dataset, where given a story from Project Gutenberg, the task is to produce a summary of the story or aspects of it based on a guiding question. The questions and summaries are original and crowdsourced; experienced writers were told to design questions that require reading significant parts of the story to answer correctly.

### 3.1.2 Question Answering

We adopt the three question answering datasets from SCROLLS (Qasper, NarrativeQA, and QuALITY), and add MuSiQue, which focuses on multi-hop question answering.

**Qasper** (Dasigi et al., 2021) contains NLP papers from the Semantic Scholar Open Research Corpus (S2ORC) (Lo et al., 2020). NLP practitioners provided questions based on the abstracts, and another set of practitioners answered given the articles.

**NarrativeQA** (Kočiský et al., 2018) contains questions and answers over books from Project Gutenberg and movie scripts from various websites. To create questions and answers, annotators were provided summaries of the books and movies

from Wikipedia, and each question was answered by one or more annotators.

**QuALITY** (Pang et al., 2022) contains stories and articles from Project Gutenberg, the Open American National Corpus, and more. Each instance contains a story and a multiple choice question; question writers were guided to write questions that require reading large portions of the story to answer correctly.

**MuSiQue** (Trivedi et al., 2022) is a multi-hop question answering dataset, where the inputs are 20 Wikipedia paragraphs and a question that requires multiple hops between different paragraphs. In the original dataset, each question also has an unanswerable twin question, where the correct answer is not present in the paragraphs. We randomly sample 100 unanswerable and 400 answerable questions for ZeroSCROLLS.

### 3.1.3 Aggregation

We create two new tasks that, by construction, require contextualizing and aggregating information from different parts of the input. Despite the inherent complexity required to solve these tasks, we design their evaluation to be simple and accurate.

**SpaceDigest** is a new sentiment aggregation task. Given 50 hotel reviews (without their ratings) from the Space dataset (Angelidis et al., 2021), the task is to determine the percentage of positive reviews. We create one example (50 reviews) per hotel from the 500 most rated hotels in the original dataset, keeping only strictly positive (rating 5 or 4) or negative (rating 2 or 1) reviews, discarding ones with an ambivalent rating of 3. To verify that humans perform this task well, we gave 5 human annotators

a shortened version of the examples (containing 10 reviews per example) and asked them to write the percentage of positive reviews. Each annotator was assigned 10 examples (100 reviews per annotator, 500 overall). The annotators aggregated their individual predictions perfectly, and had a total of 8 single-review classification errors out of the 500 reviews seen (∼98.4% accuracy).

**BookSumSort** is a new task based on the Book-Sum dataset (Kryscinski et al., 2022), which contains summaries of chapters (or parts) of novels, plays, and long poems from various sources. Given a shuffled list of chapter summaries, the task is to reorder them according to the original order of summaries in BookSum. We create the task by manually selecting the summaries of 125 books from BookSum, retaining only high-quality instances. We manually edit each summary by removing introductions, prefaces, overviews, and so forth, as well as any other information that may indicate the exact position of a summary; for example, *"Chapter 8 begins with Jane describing..."* is replaced with *"This Chapter begins with Jane describing..."* and *"As the play opens, Hippolytus announces..."* becomes *"Hippolytus announces..."*. Each list of summaries contains between 3 and 86 chapter summaries, with a median of 15 and an average of 18.8 chapters per instance. We select 4 random permutations of each list to create 500 instances.

### 3.2 Prompting

ZeroSCROLLS tests the ability to reason over long texts without any explicit training examples (zero-shot). We thus complement each data instance with an instruction that defines both the task and the desired output format (Efrat and Levy, 2020), without in-context demonstrations. While we invest effort in designing the canonical prompts for ZeroSCROLLS, the benchmark is open to further zero-shot prompt engineering (Radford et al., 2019; Schick and Schütze, 2021a,b), such as prompts that encourage chain-of-thought reasoning (Wei et al., 2022b). Table 5 contains the prompts for the summarization tasks and Table 6 contains prompts for question answering and agregation tasks.

**Prompt Structure** Figure 3 illustrates an example from the benchmark. We manually craft a prompt for each dataset, following a generic template composed of *instruction*, *context*, *query*, and *response*. The instruction describes the task, and ends with the desired output format (e.g. "Answer

---

You are given a meeting transcript and a query containing a question or instruction. Answer the query in one or more sentences.

Transcript:
User Interface: That's the same as uh on the top of it uh with the the round uh button.
Industrial Designer: Like this one.
User Interface: But uh we don't uh we don't uh disfmarker we do think it's um well disfmarker what if with ease of use, w which prefers the disfmarker which the the customer of the user prefers.
Industrial Designer: It's important. Uh I think th this is device which which has a learning curve... [The rest of the transcript is omitted]

Query:
What did the group discuss about production costs of the product?

Answer:

Figure 3: An example input in ZeroSCROLLS, taken from the QMSum dataset. The meeting transcript and the question are in *black*, and the ZeroSCROLLS prompt is in *blue*. In *copper* is a string we append to the trimmed context when the model's context window is too short to contain the entire input.

---

the query in one or more sentences." for QMSum). When the total input size is too long for a model's context window, we trim the context and append a string explicitly stating that the rest of the context is trimmed, to inform the model that it cannot see the entire context. We then concatenate the context with a header describing what kind of context it is, e.g. "Report:", "Reviews:", etc. For tasks that have queries, we append the question or query with an appropriate header. The prompt ends with a header indicating the response type (e.g. "Answer:" or "Summary:").

**Accommodations for ChatBots** Chat LLMs, such as ChatGPT and Claude, are designed to interact with humans through a chat interface. We therefore adapt our canonical prompts to accommodate these models. Specifically, omit the response header (e.g. "Summary:" or "Answer:") as it is clear, in dialogue, that the input sequence has ended. In addition, we append "Do not provide any explanation." to the instructions of question answering and aggregation tasks. For Claude, we wrap each prompt with "Human:" and "Assistant:" dialogue indicators, and for the question answering and aggregation tasks also add the instruction

to "please highlight your final answer with <{response_type}></{response_type}> tags" – as recommended by Anthropic's documentation.[3]

### 3.3 Automatic Evaluation

ZeroSCROLLS evaluation is fully automatic. Given a model's response to every test instance, we apply per-task automatic evaluation metrics. These are then averaged across tasks to produce the model's ZeroSCROLLS score. For existing datasets, we follow Shaham et al. (2022) and use the metrics provided by each dataset's authors. For our newly proposed tasks (SpaceDigest and BookSumSort), we use two new automatic metrics.

**ROUGE** *(GovReport, SummScreenFD, QMSum, SQuALITY)* ROUGE (Lin, 2004) measures ngram overlap between generated and reference summaries. For each instances, we combine ROUGE-1, ROUGE-2, and ROUGE-L into a single score by computing their geometric mean. For SQuALITY, where there are multiple references, we take the maximal value of each ROUGE type before computing the geometric mean.

**F1** *(Qasper, NarrativeQA, MuSiQue)* F1 computes unigram overlap between generated and reference answers, after normalizing white-spaces, lowercasing, omitting stopwords and punctuation (Rajpurkar et al., 2016), and transliterating any Unicode text to ASCII characters. For Qasper and NarrativeQA, where there are multiple reference answers, we take the maximal F1 score per instance.

**Accuracy** *(QuALITY)* For multiple choice questions, we compare the predicted letter (A, B, C, or D) to the reference. We use the first valid option letter surrounded by word boundaries.

**Exponential Similarity** *(SpaceDigest)* Assuming that the output is a percentage,[4] we compute the exponential similarity between the gold reference percentage $p$ and the predicted scalar $\hat{p}$:

$$ES(p, \hat{p}) = d^{-c \cdot |p - \hat{p}|}$$

We use $d = 2$ and $c = 10$, which means that, intuitively, the score gets cut by half for every 10 point deviation from the correct answer.

| Model | Params | Maximum Length | Open/Closed |
|---|---|---|---|
| T0pp | 11B | 8,192 | Open |
| Flan-T5 | 11B | 8,192 | Open |
| Flan-UL2 | 20B | 8,192 | Open |
| DaVinci003 | – | 4,096 | Closed |
| ChatGPT | – | 4,096 | Closed |
| Claude | – | 8,192 | Closed |
| GPT-4 | – | 8,192 | Closed |

Table 2: State of the art LLMs we evaluate. Exact parameter counts of closed models are not publicly available.

**Concordance Index** *(BookSumSort)* Assuming that the output is a permutation of the given chapter summary IDs,[5] we measure the amount of chapter summary pairs that are in the right order, divided by the total number of pairs $\binom{n}{2}$. The average random permutation scores 50% on this metric.

## 4 Evaluating State-of-the-Art LLMs

Using ZeroSCROLLS we conduct, to the best of our knowledge, the first systematic LLMs zero-shot performance comparison over tasks that require long text understanding.

### 4.1 Models

We evaluate both open-source models and closed products available via APIs. We apply greedy decoding to all models, and leave further research into other decoding strategies to future work. Table 2 shows the selection of models we evaluate.

**Open Source Models** We experiment with **Flan-T5-xxl** (Wei et al., 2022a) and **Flan-UL2**, the instruction-tuned versions of T5 (Raffel et al., 2020) and UL2 (Tay et al., 2023), as well as **T0pp** (Sanh et al., 2022), an LM-adapted (Lester et al., 2021) version of T5 that was finetuned on various NLP tasks for zero shot generalization. For all open-source models we use a maximum input length of 8,192 tokens (larger contexts were unstable). We also experiment with shorter context lengths and smaller variants of Flan-T5.

**Closed Models (Products)** Using product APIs, we evaluate **Claude** v1.3 from Anthropic,[6] and **DaVinci003**,[7] **ChatGPT** v0301,[8] and **GPT-4**

---

[3]https://console.anthropic.com/docs/prompt-design/classification

[4]If the output is not a percentage, we score 0%. We parse the first appearance of a percentage; e.g. for the output *"Out of 50 reviews, 20 are positive and 30 are negative, so 40% of the reviews are positive 60% are negative."* we automatically parse 40% as the answer.

[5]If the output is not a permutation, we score 0%. We discard all characters but digits, commas, and white-spaces from the output string to eliminate any prefixes such as "Order:"

[6]https://www.anthropic.com/index/introducing-claude

[7]https://platform.openai.com/docs/model-index-for-researchers

[8]https://chat.openai.com/

| Model | Tokens | GvRp $R_{geo}$ | SSFD $R_{geo}$ | QMsm $R_{geo}$ | SQAL $R_{geo}$ | Qspr F1 | Nrtv F1 | QALT AC | MuSQ F1 | SpDg ES | BkSS $C_{idx}$ | Avg |
|---|---|---|---|---|---|---|---|---|---|---|---|---|
| *Baselines* | | | | | | | | | | | | |
| Naive | - | 22.6 | 6.7 | 6.7 | 10.5 | 6.1 | 2.1 | 26.6 | 20.0 | 45.0 | 50.0 | *19.6* |
| Human | - | - | - | - | 23.6 | 67.7 | 58.2 | 93.5 | 74.8 | 93.3 | - | - |
| *Open Source Models* | | | | | | | | | | | | |
| T0pp | 8192 | 7.1 | 9.6 | 7.2 | 3.9 | 25.0 | 18.7 | 21.4 | 35.3 | 15.2 | 0.0 | *14.3* |
| Flan-T5 | 8192 | 17.6 | 7.8 | 11.0 | 8.0 | 48.3 | 19.3 | 75.2 | 46.8 | 48.7 | 16.4 | *29.9* |
| Flan-UL2 | 8192 | 16.1 | 11.5 | 13.6 | 5.7 | **56.9** | 25.5 | 75.6 | **51.3** | 36.0 | 14.0 | *30.6* |
| *Closed Models* | | | | | | | | | | | | |
| DaVinci003 | 4096 | 21.7 | 16.1 | 16.9 | 22.0 | 52.7 | 24.6 | 69.0 | 33.5 | 31.3 | 49.5 | *33.7* |
| ChatGPT | 4096 | 21.3 | 16.1 | 15.6 | 20.4 | 49.3 | 25.1 | 66.6 | 27.1 | 49.1 | 49.8 | *34.0* |
| Claude | 8000 | 24.2 | 16.1 | 14.6 | 21.0 | 52.3 | **32.6** | 84.8 | 36.1 | 61.6 | 47.4 | *39.1* |
| GPT-4 | 8192 | **26.3** | **17.3** | **18.5** | **22.6** | 50.7 | 27.6 | **89.2** | 41.1 | **62.8** | **60.5** | ***41.7*** |
| *Fine-tuned Models* | | | | | | | | | | | | |
| CoLT5 | 16384 | 41.0 | 20.0 | 22.5 | - | 53.1 | 31.0 | 47.0 | - | - | - | - |

Table 3: The ZeroSCROLLS leaderboard, at the time of writing. The dataset abbreviations stand for: GovReport, SummScreenFD, QMSum, SQuALITY, Qasper, NarrativeQA, QuALITY, MuSiQue, SpaceDigest, BookSumSort.

v0314 (OpenAI, 2023) from OpenAI. The maximal context length of these models includes both input and output.

**Task-Specific Models** To compare general-purpose LLMs (zero-shot) to task-specific models (fine-tuned), we use predictions by **CoLT5-xl** (Ainslie et al., 2023), a transformer allocating more resources to important tokens, with a maximum input length of 16,384 tokens and is the current state of the art on SCROLLS.

**Naive Baselines** We implement simple baselines for all tasks. For GovReport, SummScreenFD, QMSum, SQuALITY and NarrativeQA, we select random spans from the input document of 500, 200, 50, 120 and 4 words respectively. For Qasper, we randomly decide whether to use one of its fixed choices ("Yes", "No", "Unanswerable") or choose a random span of 15 words. For MuSiQue, we use "Unanswerable" for every instance. For QuALITY, we randomly select an option from A, B, C, or D. For SpaceDigest we always use 50%, and for BookSumSort we use the trivial permutation "$1, 2, 3, ..., n$."

**Human Performance** We provide human performance figures for 6 of the 10 tasks. For SQuALITY, Wang et al. (2022) estimate human performance by comparing one reference against the other three. Similarly, for Qasper and NarrativeQA, we calculate inter-annotator F1 on the ZeroSCROLLS subsets. We use the human scores reported by Pang et al. (2022) on the full QuALITY test set, while for MuSiQue, we combine statistics on answerable and non-answerable sets from Trivedi et al. (2022). For SpaceDigest, we use our own human annotations (Section 3.1.3) to estimate exponential similarity over 50 reviews.

## 4.2 Main Results

Table 3 shows the results for every model on every ZeroSCROLLS task, along with the average. The overall best model is GPT-4 with an average score of 41.7, and its closest competitor is Claude with 39.1, both significantly higher than the other models. We discuss the results per task category.

**Summarization** There is a clear trend where the open-source models lag behind product-grade LLMs, and that GPT-4 reaches the highest ROUGE scores on all four datasets. However, zero-shot LLMs struggle to compete with models fine-tuned per dataset (CoLT5) on those tasks, with some gap on SummScreenFd and QMSum, and a dramatic difference on GovReport (41.0 compared to 26.3). In SQuALITY, GPT-4 is only one point away from the lower bound on human performance.

**Question Answering** We see a different trend in question answering. GPT-4 achieves the best result on only one dataset, QuALITY, where it scores 89.2, close to human performance of 93.5. Flan-UL2 sets the high scores for Qasper and MuSiQue, while Claude has the best F1 on NarrativeQA, 5 points more than GPT-4. Our analysis in Section 5 reveals that GPT-4 does not conform to the required answer format, resulting in a lower score.

| Model | Tokens | GvRp $R_{geo}$ | SSFD $R_{geo}$ | QMsm $R_{geo}$ | SQAL $R_{geo}$ | Qspr F1 | Nrtv F1 | QALT AC | MuSQ F1 | SpDg ES | BkSS $C_{idx}$ | Avg |
|---|---|---|---|---|---|---|---|---|---|---|---|---|
| *Flan-T5 Across Model Sizes* | | | | | | | | | | | | |
| Flan-T5-s | 8192 | 7.6 | 4.2 | 8.3 | 3.8 | 18.5 | 11.6 | 34.6 | 21.0 | 0.0 | 0.0 | *11.0* |
| Flan-T5-b | 8192 | 5.4 | 5.1 | 9.7 | 5.6 | 14.2 | 16.5 | 48.4 | 26.9 | 0.0 | 0.3 | *13.2* |
| Flan-T5-l | 8192 | 6.9 | 6.8 | 9.7 | 5.7 | 33.6 | 20.1 | 62.4 | 33.1 | 48.0 | 0.3 | *22.7* |
| Flan-T5-xl | 8192 | 15.2 | 7.2 | 10.2 | 6.6 | 46.6 | **21.6** | 69.6 | 42.8 | 32.8 | 2.2 | *25.5* |
| Flan-T5-xxl | 8192 | **17.6** | **7.8** | **11.0** | **8.0** | 48.3 | 19.3 | **75.2** | 46.8 | 48.7 | 16.4 | *29.9* |
| *Flan-T5-xxl Across Input Lengths* | | | | | | | | | | | | |
| Flan-T5-xxl | 512 | 10.0 | 7.9 | 10.4 | 6.1 | 15.3 | 17.6 | 48.2 | 26.0 | 20.8 | 9.0 | *17.1* |
| Flan-T5-xxl | 1024 | 12.1 | 9.4 | 10.1 | 6.3 | 25.5 | 18.9 | 53.2 | 30.3 | 28.7 | 13.4 | *20.8* |
| Flan-T5-xxl | 2048 | 14.0 | **10.0** | 11.0 | 6.8 | 35.7 | 20.9 | 59.8 | 40.6 | 35.0 | 14.7 | *24.9* |
| Flan-T5-xxl | 4096 | 17.3 | 9.1 | **11.8** | 7.4 | 46.5 | **22.2** | 70.8 | **46.8** | 44.1 | 15.1 | *29.1* |
| Flan-T5-xxl | 8192 | **17.6** | 7.8 | 11.0 | **8.0** | 48.3 | 19.3 | **75.2** | **46.8** | 48.7 | 16.4 | *29.9* |
| *Claude Across Input Lengths* | | | | | | | | | | | | |
| Claude | 4096 | 23.0 | 15.0 | 14.3 | 20.2 | 47.7 | 31.7 | 76.8 | 35.8 | 61.1 | 37.6 | *36.3* |
| Claude | 8000 | **24.2** | **16.1** | **14.6** | **21.0** | 52.3 | 32.6 | 84.8 | 36.1 | 61.6 | 47.4 | *39.1* |

Table 4: Performance of Flan-T5 across model sizes, and Flan-T5 and Claude across input lengths.

**Aggregation** Our new SpaceDigest and Book-SumSort datasets enrich ZeroSCROLLS with challenges that explicitly require aggregating information across the sequence. Results indicate that both tasks are difficult for current LLMs. Performance figures for SpaceDigest show that even though sentiment analysis, counting, and divisions are all "easy" tasks for contemporary models, their combination can be quite challenging; only Claude and GPT-4 significantly outperform the naive baseline. The situation is even more dire in BookSumSort, where only GPT-4 outperforms the naive baseline.

### 4.3 Impact of Model Size and Input Length

We now discuss the effects of increasing model size (parameters) and context length (tokens). As one may expect, both dimensions improve performance on ZeroSCROLLS, suggesting that the benchmark does indeed necessitate complex reasoning over long sequences.

**Model Size** The upper section of Table 4 shows results of Flan-T5 of various sizes, ranging from S (60M parameters) to XXL (11B parameters). As expected, increasing model size drives performance upwards across almost all tasks.

**Input Length** The middle and lower sections of Table 4 show the effect of increasing the maximum number of input tokens for Flan-T5 and Claude. In general, increasing the number of tokens helps the models preform the tasks better. Claude is able to utilize the extra tokens more consistently, which results in an almost 3 point increment to its average score when going from 4k to 8k tokens. Inter-

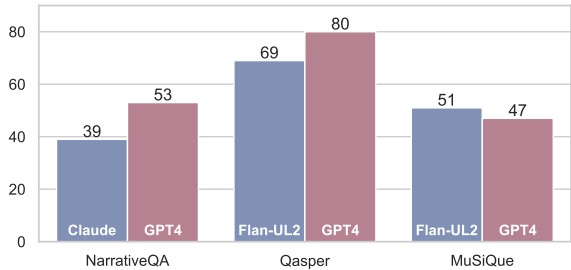

Figure 4: Human evaluation (accuracy) over 100 questions from NarrativeQA, Qasper, and MuSiQue, comparing GPT-4 to the highest scoring model of each dataset.

estingly, Flan-T5 also achieves higher scores on longer inputs in many cases, despite being trained on much shorter sequences.

## 5 Analysis

While GPT4 has the highest score on the ZeroSCROLLS leaderboard, we find it surprising that other models score higher on a number of question answering tasks. We analyze model generations and observe that GPT-4 responses do not match the desired output format (despite explicit instructions in the prompt), which results in penalization by the automatic metrics. Further analysis reveals that format discrepancy is a phenomenon that occurs across different LLMs and tasks, and is not unique to GPT-4 and question answering.

**Discrepancies in Question Answering** We analyze the responses of GPT-4 and Claude for NarrativeQA (where Claude scores 5 points higher), and the responses of GPT-4 and Flan-UL2 for Qasper and MuSiQue (where Flan-UL2 scores 6.2 and 10.2

points higher, respectively). Specifically, we sample 100 instances from each dataset, and annotate whether the answer is correct, ignoring formatting, fluency, or other factors. Figure 4 shows that, in contrast to the F1 scores, GPT-4 performs better than Claude and Flan-UL2 on NarrativeQA and Qasper, respectively, and that the gap between GPT-4 and Flan-UL2 on MuSiQue is smaller in practice.

From examining the generated texts, we learn that GPT-4 consistently generates complete answers even though the prompt instructs otherwise (see Section 3.2 and Appendix A). We further analyze 200 random instances from NarrativeQA and check whether GPT-4 and Claude respond in the specified format, i.e. "using a single phrase if possible," regardless of whether the content is correct or not. While Claude answers 191 questions in the correct format, GPT-4 does so for only 71 out of the 200 analyzed examples – explaining why GPT-4 is penalized harder by the F1 metric, despite being "correct" more often than Claude.[9]

**Format Discrepancy** Figure 5 surveys the distribution of output lengths across multiple tasks and models. In most cases, models generate outputs that fall within the distribution of reference lengths, indicating that the format criteria provided in the prompts are sufficient. However, certain task-model combinations fall outside of the reference distribution. While the NarrativeQA plot confirms our previous observation that GPT-4 generates longer answers for this task, we find that format discrepancy is not unique to this dataset or GPT-4, as different models struggle to generate texts in the correct format on different tasks; Claude generates long answers for QMSum, Flan-UL2 generates long summaries in SummScreenFD, and all models generate short summaries for GovReport, which negatively impacts their scores.

## 6   Conclusion

We introduce ZeroSCROLLS, a benchmark for zero-shot natural language understanding over long texts. ZeroSCROLLS enables systematic comparison of LLMs on tasks with naturally long input texts, and ones that require contextualizing and aggregating information from multiple documents. We evaluate

[9]Another interesting observation from analyzing NarrativeQA is that GPT-4 sometimes responds that it is unable to answer the question because the (trimmed) context does not contain the answer. It does so for 30 out of 200 cases, while Claude generates a similar response for only 5, despite both models having similar context lengths (8k).

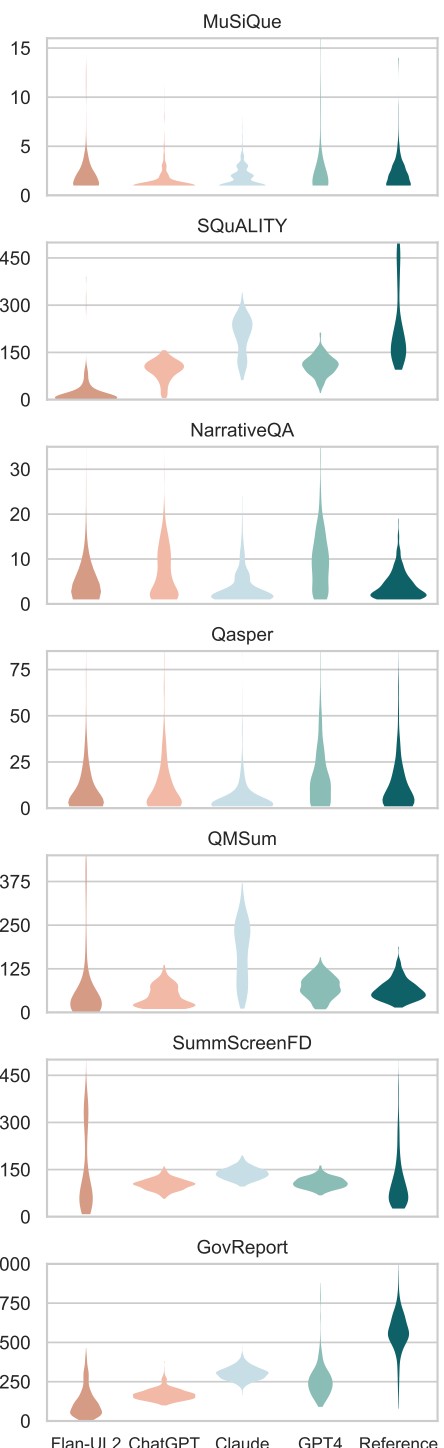

Figure 5: Distribution of the number of generated words.

open-source and production-grade LLMs to find that GPT-4 and Claude are currently the best performing models, while open-source models such as Flan-UL2 also prove powerful at long-context question answering tasks. ZeroSCROLLS remains an open challenge for LLM research, with our two

new aggregation tasks proving to be particularly difficult for contemporary LLMs.

# 7 Limitations

As language models improve, evaluating them presents a growing challenge given their ability to consistently generate coherent and reasonable text, which is harder to score, even with gold references at hand. Specifically in the zero-shot setting, where models must infer the output format from the prompt, ROUGE and F1 (ngram metrics) can assign low scores for semantically equivalent generations, with different word choices or answer lengths. Additionally, to conduct fair evaluation, we use common prompt templates across models for every task, while model-specific prompts, as well as chain-of-thought prompting may improve model performance on this benchmark. Finally, the state of the art is a moving target, and as we write these lines new long-range models, alignment methods, decoding algorithms, and prompting techniques become available; we invite researchers to evaluate their ideas on the ZeroSCROLLS leaderboard.

## Acknowledgements

This research is supported by the Yandex Initiative in Machine Learning. The benchmark is released by Tel Aviv University. All experiments were conducted by Tel Aviv University.

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

## A   Prompts

Table 5 shows ZeroSCROLLS prompts for summarization tasks, and Table 6 shows our prompts for question answering and aggregation tasks. The prompts are designed to be simple, natural, and explicit. In braces are placeholders for the text of every example.

| Task | Prompt |
|------|--------|
| GovReport | You are given a report by a government agency. Write a one-page summary of the report.

Report:
{REPORT}

Summary: |
| SummScreen | You are given a script of a TV episode. Summarize the episode in a paragraph.

Episode Script:
{SCRIPT}

Summary: |
| QMSum | You are given a meeting transcript and a query containing a question or instruction. Answer the query in one or more sentences.

Transcript:
{TRANSCRIPT}

Query:
{QUERY}

Answer: |
| SQuALITY | You are given a story and a question. Answer the question in a paragraph.

Story:
{STORY}

Question:
{QUESTION}

Answer: |

Table 5: Summarization task prompts. For chat models (ChatGPT, Claude, and GPT-4), we and omit the response header, as it is less appropriate for dialogue.

| Task | Prompt |
|---|---|
| Qasper | You are given a scientific article and a question. Answer the question as concisely as you can, using a single phrase or sentence if possible. If the question cannot be answered based on the information in the article, write "unanswerable". If the question is a yes/no question, answer "yes", "no", or "unanswerable". Do not provide any explanation.

Article:
{ARTICLE}

Question:
{QUESTION}

Answer: |
| NarrativeQa | You are given a story, which can be either a novel or a movie script, and a question. Answer the question as concisely as you can, using a single phrase if possible. Do not provide any explanation.

Story:
{STORY}

Question:
{QUESTION}

Answer: |
| QuALITY | You are provided a story and a multiple-choice question with 4 possible answers (marked by A, B, C, D). Choose the best answer by writing its corresponding letter (either A, B, C, or D). Do not provide any explanation.

Story:
{STORY}

Question and Possible Answers:
{QUESTION_AND_OPTIONS}

Answer: |
| MuSiQue | You are given several paragraphs from Wikipedia and a question. Answer the question as concisely as you can, using a single phrase if possible. If the question cannot be answered based on the information in the paragraphs, write "unanswerable". Do not provide any explanation.

Paragraphs:
{PARAGRAPHS}

Question:
{QUESTION}

Answer: |
| SpaceDigest | You are given a list of reviews about a specific hotel. Each review is either positive or negative. What is the percentage of positive reviews (e.g. 60%, 34%, etc.)? Do not provide any explanation.

Reviews:
{REVIEWS}

Percentage of Positive Reviews: |
| BookSumSort | You are given {NUM_SUMMARIES} summaries of chapters or parts of a novel, in a shuffled order, where each summary is denoted by a numerical ID (e.g. Summary 1, Summary 3, etc.). Reorder the summaries according to the original order of chapters/parts in the novel by writing a list of length {NUM_SUMMARIES} of the summary IDs (e.g. if you were given 5 summaries, one possible answer could be "5, 1, 3, 4, 2"). Do not provide any explanation.

Summaries:
{SUMMARIES}

Summary IDs in Correct Order: |

Table 6: Question answering and aggregation task prompts. For chat models (ChatGPT, Claude, and GPT-4), we add an additional instruction (in grey), and omit the response header, as it is less appropriate for dialogue.