# OpenReview forum: "ZeroSCROLLS: A Zero-Shot Benchmark for Long Text Understanding"
_EMNLP/2023/Conference — EMNLP 2023 Findings_

### Official Review · Reviewer_vUFG · 2023-08-03

**Soundness:** 4

**Excitement:**

2: Mediocre: This paper makes marginal contributions (vs non-contemporaneous work), so I would rather not see it in the conference.

**Paper Topic And Main Contributions:**

This paper introduces a new benchmark for evaluating the zero-shot performance of large language models on long text understanding tasks. The benchmark includes 10 datasets spanning summarization, question answering, and information aggregation over long texts like scientific articles, books, meeting transcripts. The authors evaluate several state-of-the-art LLMs, including Flan-T5, Claude and GPT-4. They find that GPT-4 achieves the highest overall score, but there is still significant room for improvement on certain challenging tasks like sentiment aggregation. The paper also analyzes issues like output formatting discrepancies between models and references.

**Questions For The Authors:**

See "reasons to reject" section.

**Reasons To Accept:**

1. The paper presents a comprehensive benchmark for long text understanding. The author also perform rigorous validation of their dataset construction methods. Facilitating further research in this area.

2. The overall presentation of the paper is clear and easy to follow. The claims are well supported.

**Reasons To Reject:**

1. Motivation Unclear: What is the contribution of this paper compared to the existing SCROLLS benchmark other than making it for zero-shot evaluation. What is the reason why we cannot use the test set in SCROLLS for zero-shot evaluation? If there isn't a reason, why we still want to construct a new dataset for zero-shot evaluation other than using the existing test set.

2. The paper offers limited insights other than simply benchmarking LLMs performance on long text understanding.

**Reproducibility:**

4: Could mostly reproduce the results, but there may be some variation because of sample variance or minor variations in their interpretation of the protocol or method.

**Reviewer Confidence:**

3: Pretty sure, but there's a chance I missed something. Although I have a good feel for this area in general, I did not carefully check the paper's details, e.g., the math, experimental design, or novelty.

---

> ### Author Rebuttal · Authors · 2023-08-27
>
> Thank you for your feedback!
>
> **Regarding your comments:**
>
> The rapid pace of advancement in LLMs requires that we update existing benchmarks frequently. ZeroSCROLLS is considerably more up to date than SCROLLS across three key dimensions:
> 1. While SCROLLS assumes that the model can be fine-tuned per task, ZeroSCROLLS evaluates a single model in a zero-shot setting throughout all tasks. It therefore has no training data, smaller test sets, and instruction prompts.
> 2. ZeroSCROLLS contains 4 new tasks upon SCROLLS. Of these, 2 are entirely new, and are challenging for SotA LLMs, yet are easy to accurately evaluate. Furthermore, 3 of the 10 tasks in ZeroSCROLLS involve multi-document reasoning, which was absent in SCROLLS.
> 3. ZeroSCROLLS rigorously evaluates the long-text understanding capabilities of SotA LLMs in the post-ChatGPT area.
>
> ZeroSCROLLS therefore replaces SCROLLS as the most current benchmark for testing long text understanding in state of the art models. This transition is essential especially considering the amount of new LLMs equipped with large context windows.
>
> While excitement about our work may be subjective, in practice, ZeroSCROLLS is being adopted by the research community, as evident by the increasing number of confidential submissions from prominent labs.

---

### Official Review · Reviewer_PkAz · 2023-08-04

**Typos Grammar Style And Presentation Improvements:** N/A
**Soundness:** 3

**Excitement:**

2: Mediocre: This paper makes marginal contributions (vs non-contemporaneous work), so I would rather not see it in the conference.

**Missing References:**

N/A

**Paper Topic And Main Contributions:**

This paper proposes a new benchmark ZeroSCROLLS, for natural language understanding over long texts. ZeroSCROLLS contains 10 sub-sets within 500 samples in each. Based on ZeroSCROLLS, this paper evaluates a series of large language models and compares their differences in different tasks

**Questions For The Authors:**

Question A: Are there plans to open source this benchmark in addition to submitting the model to the leader board for testing?

**Reasons To Accept:**

1. Well-written paper. The paper is easy to follow.
2. Extensive experimentation is conducted.

**Reasons To Reject:**

1. The contribution of the whole paper is limited.  It feels like this benchmark is more like a randomly selected subset of SCROLLS, and all of the datasets are from published work. The paper lacks an explanation of why these datasets were selected and what specific aspects of long-text modeling capability were evaluated. This leaves the reader unclear as to whether the benchmark is a comprehensive measure of the model's ability in the direction of long text.
2. The analysis of the experiment was insufficient. For example, order the difficulty of the selected task/dataset. At the same time, there is no more in-depth analysis of some experimental results, such as Table 4, the performance gain of FLAN-T5 tokens from 4096 to 8192 is very small or even reduced, which needs to be explained in more detail.

**Reproducibility:**

4: Could mostly reproduce the results, but there may be some variation because of sample variance or minor variations in their interpretation of the protocol or method.

**Reviewer Confidence:**

4: Quite sure. I tried to check the important points carefully. It's unlikely, though conceivable, that I missed something that should affect my ratings.

---

> ### Author Rebuttal · Authors · 2023-08-27
>
> Thank you for your feedback!
>
> **Regarding your comment about the contribution being limited:**
>
> The rapid pace of advancement in LLMs requires that we update existing benchmarks frequently. ZeroSCROLLS is considerably more up to date than SCROLLS across three key dimensions:
> 1. While SCROLLS assumes that the model can be fine-tuned per task, ZeroSCROLLS evaluates a single model in a zero-shot setting throughout all tasks. It therefore has no training data, smaller test sets, and instruction prompts.
> 2. ZeroSCROLLS is not a random subset of SCROLLS, as it contains 4 new tasks upon SCROLLS. Of these, 2 are entirely new, and are challenging for SotA LLMs, yet are easy to accurately evaluate. Furthermore, 3 of the 10 tasks in ZeroSCROLLS involve multi-document reasoning, which was absent in SCROLLS.
> 3. ZeroSCROLLS rigorously evaluates the long-text understanding capabilities of SotA LLMs in the post-ChatGPT era.
>
> ZeroSCROLLS therefore replaces SCROLLS as the most current benchmark for testing long text understanding in SotA models. This transition is essential especially considering the amount of new LLMs equipped with large context windows.
>
> While excitement about our work may be subjective, in practice, ZeroSCROLLS is being adopted by the research community, as evident by the increasing number of confidential submissions from prominent labs.
>
> **Regarding your comment about the a lack of explanation of why these datasets were selected:**
>
> Some of the tasks in ZeroSCROLLS are from the established SCROLLS benchmark, and test mainstream summarization and QA tasks over single documents across various language domains. The 4 new tasks are selected or created in this work especially to require aggregating information across the input, with 3 involving multiple documents. We will clarify this in an updated version.
>
> **Regarding your comment about the order the difficulty of the selected task/dataset:**
>
> We include estimations for human performance for 6 tasks and naive baselines for all tasks to help grasp their difficulties. For example, we find that most LLMs demonstrate impressive QA capabilities, while still struggling on our two new aggregation tasks (see 4.2).
>
> **Regarding Flan-T5 sometimes unable to capitalize 8k input tokens:**
>
> We find the result that Flan-T5 works well at all on long sequences actually quite surprising, as it was trained on sequences of only up to 512 tokens. To the best of our knowledge, we are the first to reveal such high generalization abilities of these models. In the same table you can see that a stronger model (Claude) is able to consistently benefit from the extra tokens.
>
> **Regarding your questions about publishing the data:**
>
> We will add links to all of the data, code for running experiments and calculating metrics in an updated version, the paper currently contains an anonymized link to the live leaderboard that accepts submissions. We will be happy to share the code and samples of the data in the discussions upon request.

---

### Official Review · Reviewer_Pt2m · 2023-08-12

**Soundness:** 3

**Excitement:**

2: Mediocre: This paper makes marginal contributions (vs non-contemporaneous work), so I would rather not see it in the conference.

**Paper Topic And Main Contributions:**

This paper introduces ZeroSCROLLS, a benchmark for zero-shot natural language understanding over long texts. Meanwhile, it contains two new tasks, sentiment aggregation and sorting book chapter summaries. The evaluation results among several LLMs demonstrate that long text understanding is challenging.

**Reasons To Accept:**

1. This paper introduces ZeroSCROLLS, a benchmark for zero-shot natural language understanding over long texts and evaluate the performance of several LLMs.

**Reasons To Reject:**

1. The contribution is trivial. The main contribution can be summarized that this paper extends the SCROLLS with two new tasks, sentiment aggregation and sorting book chapter summaries. More analysis and insights should be provided, such as the potential reasons of why other LLMs have poor performance among these tasks.
2. The paper should be revised carefully.
3. There is no dataset is the submission materials.

**Reproducibility:**

4: Could mostly reproduce the results, but there may be some variation because of sample variance or minor variations in their interpretation of the protocol or method.

**Reviewer Confidence:**

4: Quite sure. I tried to check the important points carefully. It's unlikely, though conceivable, that I missed something that should affect my ratings.

---

> ### Author Rebuttal · Authors · 2023-08-27
>
> Thank you for your feedback!
>
> **Regarding your comment about the contribution being trivial:**
>
> The rapid pace of advancement in LLMs requires that we update existing benchmarks frequently. ZeroSCROLLS is considerably more up to date than SCROLLS across three key dimensions:
> 1. While SCROLLS assumes that the model can be fine-tuned per task, ZeroSCROLLS evaluates a single model in a zero-shot setting throughout all tasks. It therefore has no training data, smaller test sets, and instruction prompts.
> 2. ZeroSCROLLS contains 4 new tasks upon SCROLLS, not 2. Of these, 2 are entirely new tasks, which are challenging for SotA LLMs, yet are easy to accurately evaluate. Furthermore, 3 of the 10 tasks in ZeroSCROLLS involve multi-document reasoning, which was absent in SCROLLS.
> 3. ZeroSCROLLS rigorously evaluates the long-text understanding capabilities of SotA LLMs in the post-ChatGPT era.
>
> ZeroSCROLLS therefore replaces SCROLLS as the most current benchmark for testing long text understanding in SotA models. This transition is essential especially considering the amount of new LLMs equipped with large context windows.
>
> While excitement about our work may be subjective, in practice, ZeroSCROLLS is being adopted by the research community, as evident by the increasing number of confidential submissions from prominent labs.
>
> **Regarding your comment about the analysis and insights:**
>
> We present a comprehensive empirical analysis of the performance exhibited by various LLMs. We reveal that Flan-T5 is able to perform complex tasks on sequences significantly longer than it was trained on. Furthermore, in section 5, we investigate how well LLMs adhere to formatting instructions, employing both qualitative and quantitative assessments.
>
> **Regarding your comment about the dataset not being in the submission materials:**
>
> We will add links to all of the data, code for running experiments and calculating metrics in an updated version, the paper currently contains an anonymized link to the live leaderboard that accepts submissions. We will be happy to share the code and samples of the data in the discussions upon request.

---

### Meta-Review · Area_Chair_uKZH · 2023-09-10

**Recommendation:** 4

**Metareview:**

This paper presents a new zero-shot benchmark, built upon the SCROLLS benchmark and called ZeroSCROLLS. 6 tasks are from the SCROLLS dataset (and adapted) and 4 are completely new. The objective of this new dataset is to evaluate both the zeroshot capability and the long-text understanding capabilities of SOTA LLMs.
Analysis and results are provided for 8 LLMs (3 open source models and 5 closed models) and 2 baselines for most of the tasks.
Data and code will be made available (anonymized link in the submission).
The motivation, difference with the SCROLLS benchmark and the need for such a benchmark could be better presented.

---

### Decision · Program_Chairs · 2023-10-07

**Decision:**

Accept-Findings

**Comment:**

This paper presents a new zero-shot benchmark, built upon the SCROLLS benchmark and called ZeroSCROLLS. 6 tasks are from the SCROLLS dataset (and adapted) and 4 are completely new. The objective of this new dataset is to evaluate both the zeroshot capability and the long-text understanding capabilities of SOTA LLMs.
Analysis and results are provided for 8 LLMs (3 open source models and 5 closed models) and 2 baselines for most of the tasks.
Data and code will be made available (anonymized link in the submission).
The motivation, difference with the SCROLLS benchmark and the need for such a benchmark could be better presented.